# Corner Cases:
# How Size and Position of Objects Challenge ImageNet-Trained Models

**Mishal Fatima**                                                      *mishal.fatima@uni-mannheim.de*
*University of Mannheim*

**Steffen Jung**                                                       *steffen.jung@uni-mannheim.de*
*Max Planck Institute for Informatics, Saarland Informatics Campus*
*University of Mannheim*

**Margret Keuper**                                                     *keuper@uni-mannheim.de*
*University of Mannheim*
*Max Planck Institute for Informatics, Saarland Informatics Campus*

**Reviewed on OpenReview:** *https://openreview.net/forum?id=Yqf2BhqfyZ*

## Abstract

Backgrounds in images play a major role in contributing to spurious correlations among different data points. Owing to aesthetic preferences of humans capturing the images, datasets can exhibit positional (location of the object within a given frame) and size (region-of-interest to image ratio) biases for different classes. In this paper, we show that these biases can impact how much a model relies on spurious features in the background to make its predictions. To better illustrate our findings, we propose a synthetic dataset derived from ImageNet-1k, Hard-Spurious-ImageNet, which contains images with various backgrounds, object positions, and object sizes. By evaluating the dataset on different pretrained models, we find that most models rely heavily on spurious features in the background when the region-of-interest (ROI) to image ratio is small and the object is far from the center of the image. Moreover, we also show that current methods that aim to mitigate harmful spurious features, do not take into account these factors, hence fail to achieve considerable performance gains for worst-group accuracies when the size and location of core features in an image change. The dataset and implementation code are available at `https://github.com/Mishalfatima/Corner_Cases`.

## 1 Introduction

Spurious features are defined as features that are predictive of the class label without being directly related to it. Such features are usually helpful for object recognition when the object is placed in a *perfect* environment or context. An example of that would be a sea lion near a body of water. This is because most models learn to associate water with sea lions and vice versa. On the contrary, spurious features can be extremely harmful when the object or the "core" features are observed in an unusual environment or against a spurious background. This scenario can happen when the model is deployed in the wild. Deep neural networks (DNNs) can be fooled easily to predict the label from the spurious cues in the background without relying on "object" or "core" features in the image itself. Recently, a plethora of techniques have been proposed to mitigate the reliance on unnecessary cues for image classification. Sagawa et al. (2020) introduced a distributionally robust optimization technique which, coupled with strong regularization, helped in achieving high accuracies for data groups that have strong spurious feature reliance. Similarly, Kirichenko et al. (2023) address this problem by retraining the last layer of a DNN using equal data points from different groups with core and

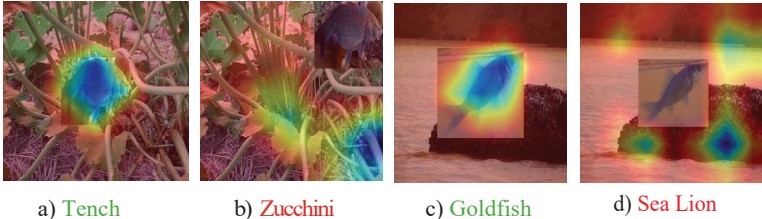

a) Tench    b) Zucchini    c) Goldfish    d) Sea Lion

Figure 1: Gradcam visualizations for Pre-trained ConvNext-Base. a) Model predicts core class *Tench* when the object is located in the center of the image, b) Spurious class *Zucchini* is predicted when the *core* class moves away from the center, c) Class *Goldfish* is predicted when the size of the core object is large ($112 \times 112$), d) Spurious class *Sea Lion* is predicted when size of core object reduces to $84 \times 84$.

spurious backgrounds. These methods are helpful when the test set exhibits similar biases as the training data, yet they fail to achieve similar performance gains when these biases are explicitly (see Figure 1).

Biases in datasets can hugely impact a deep neural network's performance. Earlier works have proven that convolutional neural networks are not entirely translation invariant and have the capacity to learn location information about objects (Biscione & Bowers, 2021). Some studies have found that models perform poorly on untrained locations (Biscione & Bowers, 2020). Similarly, object size within an input frame can lead to models performing badly when the sizes differ at inference time. The deep learning community has tried to mitigate the effect of these biases by proposing different data augmentation techniques that ensure that models are robust to changes in size and locations of the objects. However, the impact of the aforementioned factors in the presence of spurious features remains less explored.

In this work, we try to answer the questions: *In the absence of position and object size biases, how much do pre-trained models rely on spurious backgrounds to make their predictions? Are current techniques that mitigate harmful spurious features enough to tackle this problem?* Specifically, the contributions of our work are as follows:

- We calculate centeredness and size scores of different classes in ImageNet-1k (Deng et al., 2009), and analyze their relation with the level of spuriousity present in that class.

- We derive a dataset from ImageNet-1k, called **Hard-Spurious-ImageNet**, containing objects against spurious backgrounds with varying sizes and positions.

- With the help of experimentation and ablation, we conclude that the size and location of the object should be taken into account when trying to mitigate harmful spurious correlations in the dataset.

## 2 Related Work

### 2.1 Spurious Features

Moayeri et al. (2022a) show that adversarial training increases model reliance on spurious features. They also show that increased spurious feature reliance occurs when the perturbations added to core features are too small to break spurious correlations. Murali et al. (2023) show that spurious features are related with a model's learning dynamics. Specifically, "easier" features learnt in the start of model training can hurt generalization. Neuhaus et al. (2023) proposed a method to identify spurious features in the ImageNet-1k dataset and introduced a fix to mitigate a model's dependence on these features without requiring additional labels. While the proposed methods to mitigate spurious feature reliance are helpful in many cases, their efficacy is less known when factors such as size and location of core features in an image change. For example

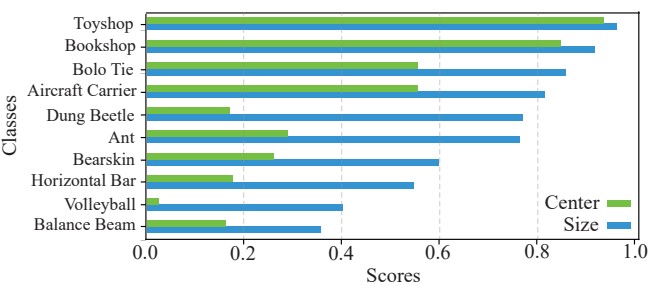 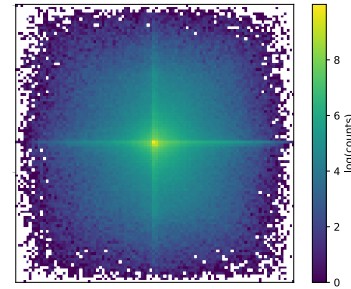

Figure 2: **Left**: ImageNet-1k classes and their center and size scores. *Toyshop* has largest center and size scores, whereas *Volleyball* has smallest center score and *Balance Beam* has smallest size score. Other classes are sampled randomly for visualization. **Right**: Counts in log scale of relative centers of ground truth bounding boxes containing the object corresponding to the image class (ImageNet-1k validation set). Most object centers are concentrated around the image center, while some are present along the main axes. Objects of interest are rarely present in image corners.

Alshami et al. (2025) propose self-supervised feature masking to reduce model reliance spurious features. Prasse et al. (2025) propose localized concepts in concept bottleneck models, making reliance on spurious features inspectable.

## 2.2 Existing Datasets

To study distribution shift robustness, Sagawa et al. (2020) introduced the Waterbirds dataset, constructed by compositing bird images from CUB-200-2011 (Wah et al., 2011) onto backgrounds from Places (Zhou et al., 2018) to induce a spurious correlation between bird species and scene type. Its test split includes counter-correlated "worst-group" examples that challenge models relying on spurious background features. Xiao et al. (2021) investigate how strongly deep learning models depend on background cues, often spurious correlations, when classifying images in ImageNet-1k. Their central question is whether models truly recognize the object itself or are misled by its surrounding context. To explore this, they introduce ImageNet-9, a reduced version of ImageNet-1k containing nine broad classes (e.g., dog, bird, vehicle). Using bounding boxes, they separate foregrounds from backgrounds and then recombine them in various controlled ways to test model sensitivity. Our work builds on these ideas but differs in scope: we use the full set of ImageNet-1k classes and focus on how spurious background correlations interact with object size and spatial location.

Moayeri et al. (2022b) propose a dataset derived from ImageNet-1k with segmentation masks for a subset of images. These masks label entire objects and various visual attributes. They name this dataset RIVAL10 and also test different models' sensitivity to noise in backgrounds and foregrounds. Moayeri et al. (2022c) propose a dataset with segmentation masks for images in 15 classes of ImageNet-1k. These images have high spurious features. They attribute this to objects being small and less centered in these images. Singla & Feizi (2022) label spurious and core features for ImageNet-1k samples. They achieve this by making use of activation maps as soft masks. Moayeri et al. (2023) rank images in ImageNet-1k dataset based on spurious cues present. They show that spurious feature reliance is influenced more by the data a model is trained on rather than how a model is trained. Lynch et al. (2025) propose a photo-realistic dataset with many-to-many spurious correlations between different groups of spurious attributes and classes. One work closely related to ours is that of Yung et al. (2022). They do a fine-grained analysis of the robustness of different models by varying factors such as object size, location, and rotation. Our technical contributions differ from theirs because we take into account the spuriousity level of backgrounds and correlate it with the above factors as well.

## 2.3 Biases in Datasets

While capturing images through a camera, humans often tend to place the region of interest in the center. Due to this, there often exists a bias in classification datasets where objects are mostly located in the center

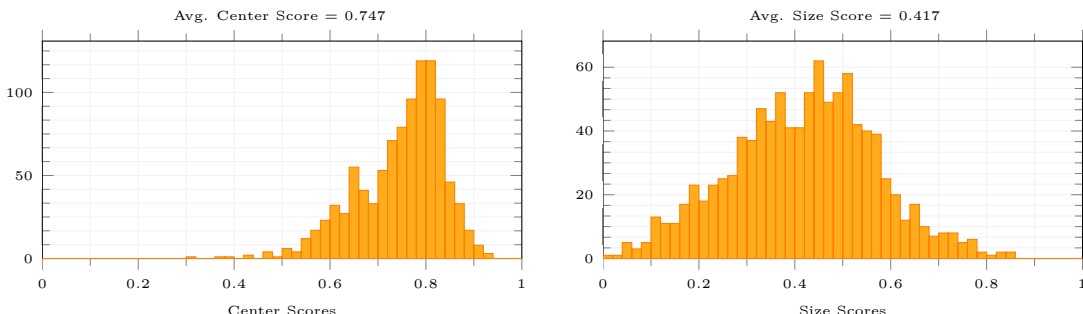

Figure 3: Histograms showing distribution of scores in different classes of ImageNet-1k validation dataset.

of images and away from the boundary of the image. Exploiting the center bias in ImageNet-1k, resizing and center cropping has been usually used for testing image classification models. Taesiri et al. (2024) show that there exists a strong center bias in out-of-distribution benchmarks such as ImageNet-A (Hendrycks et al., 2021) and ObjectNet (Barbu et al., 2019) by using resize and center crop operations only. They resize the image to multiple scales and patchify it, followed by a center crop operation at every patch. Doing this, they end up with different zoomed-in versions of the input images. The computed accuracy of the center crop is maximum showing the presence of a strong center bias in the dataset. In this paper, we do an in-depth analysis of the presence of center and size bias in every class of ImageNet-1k by computing distinct scores. The detailed explanations of these scores are given in following sections.

## 3 Biases in ImageNet

In this section, we quantitatively analyze positional and size biases present in ImageNet-1k. To get a better sense of these biases, we propose *centeredness* and *size* scores. It is important to note that these scores are calculated by the provided ground truth bounding box of ImageNet-1k.

### 3.1 Centeredness Score

In the majority of images in ImageNet-1k, the objects of interest are located in the image's center. Hence, in this paper, we use "positional" and "center" as synonyms. To understand the extent of center bias prevalent in ImageNet-1k, we propose a *Center Score* defined as

$$C_c = \frac{1}{M}\frac{1}{N}\sum_{i=1}^{M}\sum_{j=1}^{N} 1 - (\|I_{i,c} - O_{i,j,c}\|_\infty), \tag{1}$$

where $C_c$ is the centeredness score for class $c$, $M$ is total number of images in the class, $N$ is total number of objects within a frame, $I$ is image center, and $O$ is object center. The distance between image center and object center is calculated by the $\ell_\infty$ norm. We choose this norm based on the fact that images are rectangular grids and the distances are measured in horizontal and vertical terms, not diagonally. It is subtracted from 1 to establish a direct relationship between the score and center bias prevalent in the class $c$.

### 3.2 Size Score

To measure the average sizes of objects within images, we define a size score as

$$S_c = \frac{1}{M}\frac{1}{N}\sum_{i=1}^{M}\sum_{j=1}^{N} \frac{h_j w_j}{H_i W_i}, \tag{2}$$

where $S_c$ is the size score for class $c$, $h$ and $w$ refer to the height and width of object $j$ in image $i$. $H$ and $W$ are the height and width of the image itself. Figure 2 (left) shows the center and size scores of different

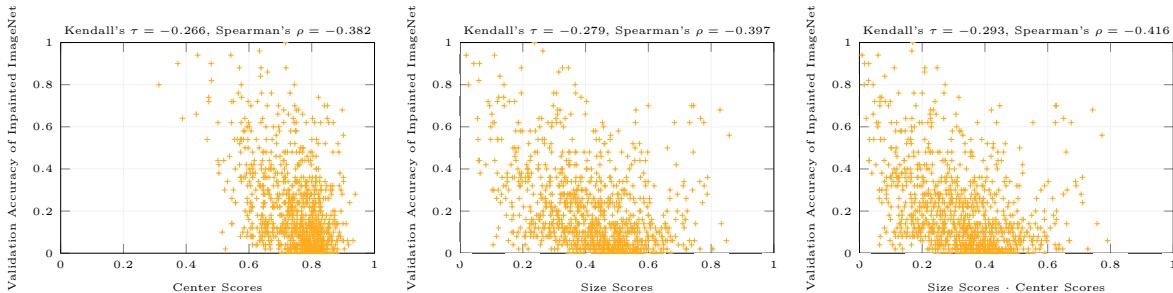

Figure 4: Correlation between the validation accuracy on inpainted ImageNet (as explained in Subsection 3.3) and, from left to right, center scores, size scores, and their product, respectively. Jointly considering center and size score shows strongest negative correlation with the accuracy.

classes, with *Toyshop* having the maximum center and size scores. The histograms in Figure 3 show the distribution of center and size scores of all the classes in the ImageNet-1k validation data. It can be seen that the majority of the classes in ImageNet-1k are highly centered with objects of interest occupying half of the image pixels on average. These scores are calculated by using Ground Truth bounding boxes of ImageNet-1k.

### 3.3   Relationship with the Level of Spuriosity

To establish a correlation between centeredness and size scores of every class to spurious feature reliance in ImageNet-1k, we first calculate the validation accuracies of different classes in ImageNet-1k with object information removed. We achieve this by using Inpaint-Anything (Yu et al., 2023) with the goal of creating a more realistic effect when the region of interest is removed from the image. The input to Inpaint Anything are the object bounding boxes and it makes use of Segment Anything (SAM) (Kirillov et al., 2023) to predict masks for objects within these bounding boxes. These predicted masks are then input to the inpainting model LaMa (Suvorov et al., 2022) which fills the masked region predicted by SAM. Finally, we resize the inpainted images to $224 \times 224$. We use ConvNext-Base (Liu et al., 2022) pre-trained on ImageNet-22k (Russakovsky et al., 2015) and fine-tuned on ImageNet-1k, to compute the validation accuracies for the inpainted dataset. Classes with higher validation accuracies indicate higher spurious feature reliance, since the model has learnt to associate the class label not just with the core object, but also with the background information. In order to assess the correlation present between center and size scores and the level of spuriousity present in different classes of ImageNet-1k, we use Kendall's $\tau$ coefficient and Spearman's correlation coefficient. The negative correlation values (see Figure 4) depict that there is an inverse relationship between both inpainted data's accuracy and the different considered scores, which validates the hypothesis that a higher spurious feature reliance is observed in case of non-centered large object sizes. The correlation is overall rather weak, which is to be expected since different classes are differently hard to classify, even from their core features.

## 4   Hard-Spurious-ImageNet

Similar to the waterbirds dataset (Sagawa et al., 2020), we say that every datapoint $(x, y)$ has an attribute $a(x) \in A$ which is spuriously correlated with label $y$. We conjecture that the strength of the correlation between attribute $a(x)$ and label $y$ is controlled by two factors: size $s$ and position $p$ of the core features in the input image. To this end, we propose **Hard-Spurious-ImageNet**, a set of synthetic datasets to illustrate the problem of spurious feature reliance in the presence of varying object bounding box sizes, locations, and backgrounds. The prime motivation of creating the dataset is to have precise control over these factors and help the community build robust models against stronger spurious cues. We consider the image content within the provided ground truth object bounding boxes for ImageNet-1k as core features and the features outside the bounding box as the background. In ImageNet-1k, bounding boxes are available for all images in the validation data, yet only a subset of images in training data are annotated. The images are annotated

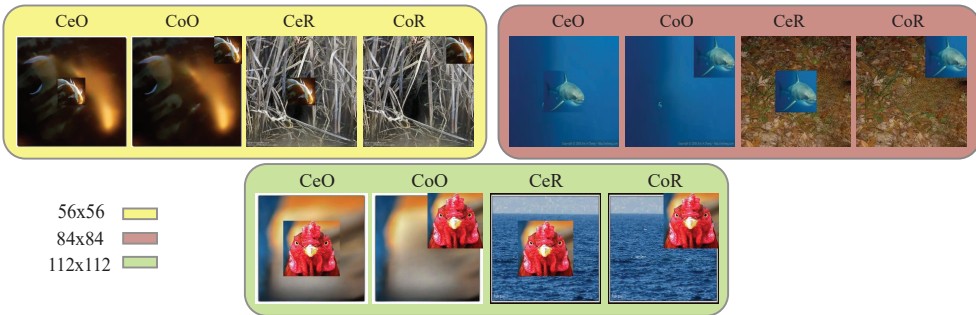

Figure 5: Different samples from Hard-Spurious-ImageNet. Image size remains same in all images, i.e. $224 \times 224$, whereas object size changes. Label of every image is same as foreground object.

and verified through Amazon Mechanical Turk. These annotations provide an estimate of the location of core features in any image. As a first step, we want to disentangle core features from the rest of the image. We achieve this by cropping out the core objects from the images and inpainting the resulting image, as explained in the previous section. Next, we resize core object bounding boxes to different sizes, and place them in two different locations against inpainted backgrounds. The size and location of core objects and the kind of background chosen, gives rise to different groups in the (see Figure 5). To efficiently gauge the performance of these different groups, we categorize them as follows:

- **Group CeO**: Core object in the **Ce**nter of image against its **O**riginal inpainted background.

- **Group CoO**: Core object in the top right **Co**rner of image against its **O**riginal inpainted background.

- **Group CeR**: Core object in the **Ce**nter of image against **R**andom inpainted background.

- **Group CoR**: Core object in the top right **Co**rner of image against **R**andom inpainted background.

We consider three core object sizes: $56 \times 56$, $84 \times 84$, and $112 \times 112$. It is important to note that all the inpainted backgrounds have already been resized to $224 \times 224$, so the core object sizes mentioned above represent $\frac{4}{64}$th, $\frac{9}{64}$th, and $\frac{16}{64}$th of the whole image. We also experimented with object masks obtained from the Segment Anything (Kirillov et al., 2023) model rather than the provided bounding boxes as foreground objects (see Table 9 in supplementary). We observed that the mask quality obtained from Segment Anything model for some objects was not good enough, hence, we used provided bounding boxes for this work.

Furthermore, we also modify the test set by randomly positioning the object in the image (any random corner instead of top-right) and preserving the aspect ratio (AR) of the object by resizing the shortest size to 56, 84, and 112, while maintaining the AR. In cases where the resized size exceeds the image canvas ($224 \times 224$), it is downsampled such that the longer size is 224 and the shorter size is as close as possible to the target size. The above grouping for the AR preserving variant changes as follows.

- **Group CeO**: Core object resized to preserve aspect ratio placed in the **Ce**nter of image against its **O**riginal inpainted background.

- **Group CoO**: Core object resized to preserve aspect ratio placed in any random **Co**rner of image against its **O**riginal inpainted background.

- **Group CeR**: Core object resized to preserve aspect ratio placed in the **Ce**nter of image against **R**andom inpainted background.

- **Group CoR**: Core object resized to preserve aspect ratio placed in any random **Co**rner of image against **R**andom inpainted background.

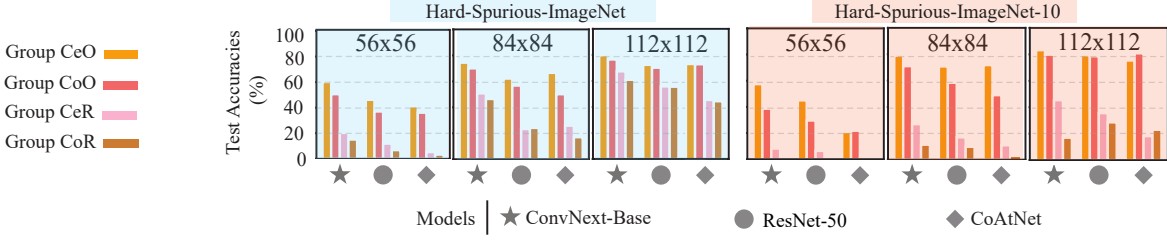

Figure 6: Benchmarking results of different models on Groups CeO (Core object in Center against its Original background), CoO (Core object in top right Corner against its Original background), CeR (Core object in Center against Random background), and group CoR (Core object in top right Corner against a Random background). Performance for our Hard-Spurious-ImageNet-10 is the worst across all groups.

We report results on both test sets. The results for Hard-Spurios-ImageNet and its AR-preserving variant are given in Table 1. We test the performance of the datasets on 5 different pre-trained architectures: ConvNext-Base (Liu et al., 2022), ResNet-50 (He et al., 2016), CoATNet (Dai et al., 2021), Hiera-Base with MAE (Ryali et al., 2023), and MViTv2-small (Li et al., 2022). All models are pretrained on ImageNet-1k only. Across all models, the performance on Group CoR for size $56 \times 56$ is the worst.

### 4.1 Hard-Spurious-ImageNet-10

Randomly chosen backgrounds have varying levels of spuriousity based on the classes they are taken from. We derive a variant of the proposed dataset where, instead of choosing backgrounds in a random fashion, they are chosen based on the level of spurious features present in them. To achieve this, we first analyze the level of spuriousity present in every class. We give inpainted images without the core objects, as input to the pretrained ConvNext-Base model , and record the accuracies of every class. The classes where accuracies are high indicate that the model has learnt to predict the class label without the presence of core objects. On the contrary, classes for which the accuracy is low are highly reliant on core features to make predictions. We choose 10 classes that are highly spurious, namely: *snorkel, bobsled, maypole, potter's wheel, gondola, bearskin, volleyball, basketball, canoe, geyser, and yellow lady's slipper* as backgrounds. For foreground objects, we choose 10 classes with high reliance on core features such as: *bluetick, box turtle, Chihuahua, Japanese spaniel, Maltese dog, Shih-Tzu, Blenheim spaniel, papillon, Rhodesian ridgeback, and basset*. We combine the above-mentioned foregrounds and backgrounds to create a dataset with 10 classes of foreground objects and highly spurious backgrounds. Similar to before, for every class, the chosen background class remains same for all images belonging to that class, but the backgrounds can differ from one image to another. Finally, we create four groups for the dataset and test on pre-trained models.

## 5 Experimental Results

We test the robustness of different models with Hard-Spurious-ImageNet and its variants. The results with Hard-Spurious-ImageNet-10 are given in the supplementary (see Table 8). Images are normalized with mean and standard deviation of the ImageNet-1k dataset. We use HuggingFace PyTorch models to test the dataset.

Figure 6 shows test accuracies of the proposed data on three pretrained models. ConvNext-Base performs best across all CNN architectures, whereas Hiera performs better compared with MViTv2 model. The difference in accuracy between groups CeR and CoR, when the core object size is $112 \times 112$ is less across all the models. This indicates that the core feature size is big enough for the model to ignore changes in location. Moreover, $\frac{1}{4}$th of the number of pixels in the image are occupied by core features in this case, so backgrounds are less exposed as compared to when the core object size is even less. Another interesting observation is that the impact of size change is far stronger on model performance than the location of core features. We also observe that in almost all the groups, there is significant drop in performance compared with clean accuracies on the standard validation dataset (see Table 1). We also see that the pretrained models perform better on aspect

Table 1: Comparison of group test accuracies on Hard-Spurious-ImageNet between original and aspect-ratio-preserving test sets. Scores are shown as *Original / AR-Preserving*. Pretrained models consistently perform better on the AR-preserving variant.

| Model | Clean Accuracy | Object Resolution | Group Accuracies (Original / AR Preserving) | | | |
|---|---|---|---|---|---|---|
| | | | CeO | CoO | CeR | CoR |
| ConvNeXt-Base | 84.43 | 56 | 49.69 / **58.56** | 45.57 / **56.03** | 7.42 / **16.03** | 7.12 / **18.42** |
| | | 84 | 53.24 / **67.24** | 62.17 / **70.34** | 4.27 / **19.85** | 22.38 / **40.36** |
| | | 112 | 73.20 / **78.35** | 71.21 / **77.91** | 32.27 / **50.34** | 36.53 / **59.79** |
| ResNet-50 | 81.21 | 56 | 47.17 / **57.77** | 42.39 / **54.45** | 15.15 / **31.11** | 11.11 / **29.20** |
| | | 84 | 63.87 / **71.02** | 61.50 / **69.64** | 39.81 / **54.04** | 36.16 / **54.22** |
| | | 112 | 71.76 / **76.22** | 70.57 / **76.06** | 55.90 / **65.77** | 55.53 / **67.43** |
| CoATNet | 82.39 | 56 | 40.59 / **52.32** | 37.66 / **48.56** | 6.80 / **17.71** | 4.61 / **15.26** |
| | | 84 | 64.78 / **71.54** | 50.99 / **65.01** | 27.92 / **44.18** | 20.30 / **38.40** |
| | | 112 | 72.25 / **77.73** | 72.08 / **76.84** | 46.30 / **61.78** | 47.64 / **61.49** |
| Hiera | 84.49 | 56 | 50.08 / **60.24** | 35.66 / **48.57** | 4.87 / **15.71** | 1.36 / **8.17** |
| | | 84 | 68.10 / **74.63** | 56.54 / **66.74** | 21.81 / **41.25** | 12.49 / **30.58** |
| | | 112 | 74.49 / **80.14** | 69.89 / **76.92** | 47.68 / **64.54** | 38.05 / **56.16** |
| MVitv2 | 83.77 | 56 | 42.10 / **54.35** | 32.07 / **45.83** | 5.37 / **16.97** | 1.54 / **10.43** |
| | | 84 | 67.84 / **73.37** | 51.56 / **64.36** | 30.34 / **44.98** | 14.18 / **32.78** |
| | | 112 | 70.53 / **78.10** | 65.15 / **74.93** | 47.83 / **63.58** | 37.72 / **56.51** |

ratio preserving variant of the dataset compared to the original dataset. This indicates that in the presence of resolution related artifacts, it becomes more difficult to classify the core object correctly.

Based on the above observations, we divide all the 12 groups consisting of different core feature sizes and locations into three distinct categories: **Easy**: This set consists of Groups CeO and CoO for larger core feature sizes, i.e. $84 \times 84$ and $112 \times 112$, as these groups seem to be doing considerably better than the rest. **Hard**: Groups CeR and CoR are the worst performing across all architecture for core feature sizes $56 \times 56$ and $84 \times 84$. We categorize them as **Hard** group. The remaining groups, i.e. groups CeO and CoO for size $56 \times 56$, and groups CeR and CoR for size $112 \times 112$ seem to be performing moderately, we put them in **Medium** category.

Following the analysis done earlier (see Figure 3), we find that most of the images in ImageNet-1k are centered with an estimated size score of $\approx 0.5$, indicating that on average, the core features in an image occupy half the number of pixels of the entire image. Keeping this in mind, we create the training data of Hard-Spurious-ImageNet consisting of majority and minority groups, where the number of images belonging to majority groups are far more than in minority groups. This is done to replicate the long-tailed distribution nature of the ImageNet-1k dataset in terms of hardness. For the training data, we consider 80 images per group in the Easy category and 10 images from groups in Medium and Hard categories. This brings the total to 400 images per class in the training data. Out of the 400 images, 320 images belong to the Easy group and 80 to the Medium and Hard groups. For the validation set, we use a balanced dataset having equal data points from every group. We use 20 images per group, resulting in 240 images per class. Both training and validation set of Hard-Spurious-ImageNet are derived from training data of ImageNet-1k, whereas the test set is derived from the validation data. The test set is also balanced, comprising 50 images per group, totaling 600 images in every class. The details of the dataset can be found in Table 11 in the supplementary.

## 5.1 Vision Language Models

We do zero-shot classification with EVA-Giant (Sun et al., 2023) and CLIP (Radford et al., 2021) models. EVA-CLIP was pretrained on LAION-400M (Schuhmann et al., 2021) with CLIP and fine-tuned on ImageNet-1k. The results (see Table 2) indicate improved performance across all groups, demonstrating the model's effectiveness. Similarly, we also report results on CLIP with ViT-B/32 backbone (trained on LAION-400M)

Table 2: Comparison of group accuracies for CLIP and EVA-CLIP models under original and AR-preserving variants. Each cell shows original / AR value. AR-preserving improves performance across all groups.

| Model | Clean Accuracy | Object Resolution | Group Accuracies (Original / AR Preserving) | | | |
|-------|----------------|-------------------|------|------|------|------|
| | | | CeO | CoO | CeR | CoR |
| CLIP | 59.25 | 56 | 29.81 / **39.83** | 22.92 / **31.58** | 7.51 / **16.29** | 4.18 / **10.99** |
| | | 84 | 43.06 / **52.02** | 35.68 / **45.97** | 20.65 / **32.14** | 14.63 / **26.11** |
| | | 112 | 49.53 / **57.35** | 45.57 / **54.42** | 30.47 / **42.06** | 25.95 / **38.23** |
| EVA-CLIP | 88.87 | 56 | 72.43 / **78.58** | 62.13 / **72.31** | 43.09 / **57.61** | 26.79 / **45.70** |
| | | 84 | 81.88 / **85.22** | 78.61 / **83.11** | 65.77 / **73.35** | 58.38 / **68.71** |
| | | 112 | 85.54 / **87.27** | 84.03 / **86.27** | 76.65 / **80.33** | 72.32 / **77.45** |

Table 3: Group accuracies for Vision Transformer models under original and AR-preserving variants. Each cell shows original / AR value. Performance increases both with model scale and with AR preservation.

| Model | Clean Accuracy | Object Resolution | Group Accuracies (Original / AR Preserving) | | | |
|-------|----------------|-------------------|------|------|------|------|
| | | | CeO | CoO | CeR | CoR |
| ViT-Tiny | 75.44 | 56 | 35.91 / **45.58** | 27.52 / **37.27** | 3.41 / **10.87** | 1.83 / **7.66** |
| | | 84 | 53.10 / **61.94** | 46.40 / **57.07** | 15.66 / **31.13** | 12.68 / **27.54** |
| | | 112 | 62.98 / **69.74** | 60.04 / **67.26** | 32.41 / **47.94** | 30.96 / **46.17** |
| ViT-Small | 81.37 | 56 | 46.59 / **57.51** | 36.25 / **48.36** | 8.49 / **21.93** | 4.63 / **15.67** |
| | | 84 | 63.85 / **71.86** | 57.73 / **67.60** | 30.61 / **47.51** | 25.49 / **43.03** |
| | | 112 | 72.48 / **77.62** | 69.36 / **75.56** | 50.33 / **62.65** | 46.62 / **60.61** |
| ViT-Large | 85.83 | 56 | 59.33 / **69.13** | 48.38 / **60.64** | 19.61 / **37.04** | 11.09 / **27.02** |
| | | 84 | 74.44 / **80.04** | 69.49 / **76.52** | 47.32 / **61.37** | 41.18 / **56.02** |
| | | 112 | 80.54 / **83.58** | 78.19 / **82.18** | 64.85 / **72.74** | 60.11 / **70.35** |

by giving the model an image and specifying all class names as a prompt. The results with EVA show that the performance is much better on hard groups, showing the strength of the model, although the overall trend still persists. Also, EVA is fine-tuned with ImageNet-1k - whereas CLIP is not which also explains the gap in accuracy and sheds light on the importance of ImageNet pretaining for the proposed dataset. Performance across AR-preserving variant is better than the original dataset.

## 5.2 Scalability Analysis

In order to do a scalability analysis to check whether large-scale models perform better or not, we take a series of models for ViT (Dosovitskiy et al., 2021) increasing in the order of scale (all trained on ImageNet-21k (Russakovsky et al., 2015) and fine-tuned on ImageNet-1k), and evaluate the test set of the proposed dataset and it AR-preserving variant (see Table 3). ViT-Tiny has 5.7M parameters, ViT-Small has 22M, and ViT-Large has 307M parameters. As expected, we see that large models perform better across all groups.

## 5.3 Training Strategy of Vision Transformer Models

In order to access which training strategy is best suited for this task, we report results from two ViTs, all fine-tuned on ImageNet-1k and roughly having the same parameter count, i.e., 22M. We note that across images with the original background, the difference in performance across both models is minimal, whereas across groups with random backgrounds, ViT-Small performs better, showing that knowledge distillation in DeiT (Touvron et al., 2021) not making a huge difference in the classification (see Table 4).

The above results indicate that larger models perform better on the challenging benchmark. Training strategy like knowledge distillation in ViTs is not very useful. VMLs that have been fine-tuned on ImageNet-1k (e.g. EVA) perform the best on the dataset, even for the challenging group CoR, compared to all other models. We also see that if not fintuned on ImageNet-1k specifically (see Table 2), we see a sharp drop in performance, as is evident in case of CLIP. We also see there exists a sharp difference in performance between clean accuracies

Table 4: Group accuracies for DeiT-Small and ViT-Small models under original and AR-preserving variants. Each cell shows accuracy as *original / AR-preserving.* AR-preserving improves performance across all groups.

| Model | Clean Accuracy | Object Resolution | Group Accuracies (Original / AR Preserving) | | | |
|---|---|---|---|---|---|---|
| | | | CeO | CoO | CeR | CoR |
| DeiT-Small | 79.85 | 56 | 41.78 / **52.81** | 28.86 / **42.00** | 3.53 / **13.72** | 1.08 / **7.52** |
| | | 84 | 62.73 / **69.33** | 47.31 / **60.98** | 28.08 / **42.30** | 10.16 / **27.89** |
| | | 112 | 68.02 / **75.09** | 67.60 / **73.12** | 44.12 / **60.29** | 38.85 / **54.56** |
| ViT-Small | 78.84 | 56 | 42.42 / **52.40** | 33.53 / **44.99** | 7.86 / **19.87** | 4.82 / **15.88** |
| | | 84 | 59.13 / **67.46** | 54.31 / **63.84** | 27.57 / **43.81** | 25.54 / **42.39** |
| | | 112 | 68.81 / **74.21** | 66.35 / **72.30** | 47.44 / **59.33** | 46.19 / **58.84** |

Table 5: ResNet-50 baseline versus augmented models on original and aspect ratio (AR) preserving variants of the dataset. Values are shown as *Original / AR Preserving* within each model variant. Aspect ratio preserving improves performance across most groups, while data augmentation boosts overall accuracy.

| Model | Clean Accuracy | Object Resolution | Group Accuracies (Original / AR Preserving) | | | |
|---|---|---|---|---|---|---|
| | | | CeO | CoO | CeR | CoR |
| ResNet-50 Baseline | 76.14 | 56 | 38.79 / **49.78** | 24.59 / **36.10** | 10.43 / **22.78** | 3.28 / **12.57** |
| | | 84 | 56.47 / **65.07** | 45.91 / **57.50** | 31.78 / **46.17** | 20.74 / **37.59** |
| | | 112 | 65.72 / **71.26** | 60.18 / **67.55** | 49.28 / **59.59** | 41.96 / **54.78** |
| ResNet-50 Augmented | 80.84 | 56 | 46.56 / **58.00** | 34.35 / **47.70** | 10.76 / **27.99** | 2.93 / **16.93** |
| | | 84 | 64.09 / **72.17** | 55.24 / **66.26** | 39.68 / **55.06** | 16.59 / **42.37** |
| | | 112 | 71.55 / **76.73** | 66.61 / **74.43** | 42.30 / **61.67** | 35.80 / **60.47** |

and group accuracies, indicating the challenging nature of the benchmark. Although the EVA-CLIP model minimizes this difference across many groups, the overall trend persists.

## 5.4 Effects of Data Augmentation

To measure the effect of data augmentations, we compared vanilla ResNet-50 trained without any augmentations on ImageNet-1k with an advanced training recipe involving auto-augment (Cubuk et al., 2019), random erase (Zhong et al., 2020), mixup (Zhang et al., 2018), and CutMix (Yun et al., 2019). The results (shown in Table 5) indicate that while data augmentation increases accuracy across groups CeO, CoO, and CeR, the performance decreases in case of group CoR for all sizes. This indicates that standard data augmentation approaches do not take into account the presence of spurious features in the data while augmenting, hence, may end up highlighting them instead. Moreover, the gap in performance still persists across all four groups for a given core object size. This hints that mere data augmentation strategies are insufficient to deal with this problem. It should be noted that clean accuracies differ here slightly because the training recipes in both models are slightly different than what is reported for benchmark.

## 5.5 Group Robustness Methods

Since the proposed original dataset is hardest of the two variants (original/AR-preserving), we measure the performance of the original dataset using simple fine-tuning and two state-of-the-art group robustness methods. Empirical Risk Minimization or **ERM** (Vapnik, 1991) is the standard paradigm for training machine learning models. Under ERM, models are optimized to minimize the average loss across the training distribution, typically leading to high overall accuracy on datasets drawn from that distribution, yet it often fails in real-world settings where spurious correlations or group imbalances exist. Deep Feature Reweighting (**DFR**) (Kirichenko et al., 2023) is a debiasing method that addresses spurious correlations by retraining only the last (classification) layer of a pre-trained model. This retraining is done using a balanced subset of the training data, where each group (e.g., defined by some spurious attribute) is equally represented. The intuition is that earlier layers have already learned generalizable features, and reweighting the last layer with group-balanced data helps reduce reliance on spurious signals without requiring full retraining. Just Train Twice (**JTT**) (Liu et al., 2021) is a two-stage method that aims to mitigate bias by identifying and focusing

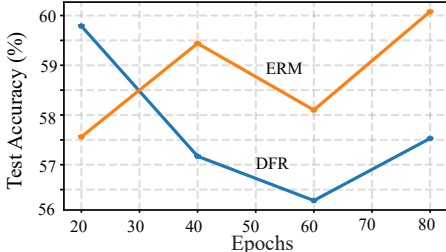 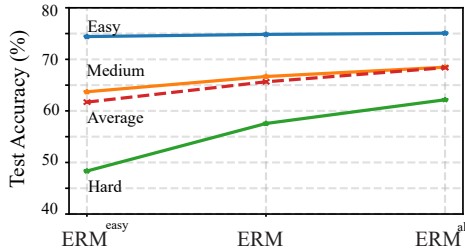

Figure 7: (**left**) The effect of training epochs of ERM model on the performance of DFR. ERM model trained with 20 epochs gives the highest performance for DFR. (**right**) ERM$_{all}$ narrows the gap between easy, medium, and hard groups.

on "hard" examples. First, a standard ERM model is trained. Then, misclassified examples from this model are upsampled in the training data by a factor $\lambda_{up}$, under the assumption that these examples are more likely to belong to underrepresented or challenging groups. A second model is trained on this reweighted dataset, helping it to better handle these minority or spurious-prone cases. We experiment with different variations of the above methods.

## 5.6 Implementation Details

We use pretrained ResNet-50 trained on ImageNet-1k for our experiments. The Base model (pretrained ResNet-50) is fine-tuned with batch size 256, constant learning rate of 0.001 for 20 epochs. The input images are randomly cropped with an aspect ratio in the bounds (0.75,1.33) and finally resized to $224 \times 224$. Horizontal flipping is applied afterward. A momentum of 0.9 and weight decay of 0.001 is used. For DFR, we normalize the embeddings using mean and standard deviation of validation data used to train the last layer, and use the same statistics to normalize embeddings of test data. We re-train the last layer for 1000 epochs, learning rate of 1, cosine learning rate scheduler and SGD optimizer with full-batch. We use $\ell_2$ regularization with $\lambda$ set to 100. These hyperparamters are similar to the ones set by Kirichenko et al. (2023) for optimizing the last layer for ImageNet-9 dataset (Xiao et al., 2021). Since, the data distribution in the proposed dataset and ImageNet-9 is similar, we assumed the same hyperparameters. In case of JTT, models have the same hyperparameters as the ERM trained model. $\lambda_{up}$ is set to 50.

## 5.7 Results

The results in Table 6 show that pretrained ImageNet models perform worst on the hard group. This could be attributed to the fact that the model has very little exposure to small core features against spurious backgrounds in the training data. The ERM model does better across easy, medium and hard groups, but there still exists a disparity in performance among the three groups.

DFR is able to perform slightly better in the Hard group by sacrificing some accuracy in Easy and Medium groups. The average test accuracy is similar for ERM and DFR. The performance with JTT also decreases, which hints that the task of learning data has become difficult for the model in the presence of upsampled images. Since the embeddings in DFR are dependent on the ERM-trained model, we also analyze how the number of training epochs the ERM model is trained for, impacts the DFR performance. The epochs for retraining the last layer remain fixed to 1000, all other hyperparameters also remain the same for DFR models trained with different ERM-trained embeddings. The left plot in Figure 7 indicates that, when the base model is fine-tuned for 20 epochs, the performance of DFR on the test set increases. As the training time increases for ERM, performance by DFR decreases, whereas the ERM model continues to improve.

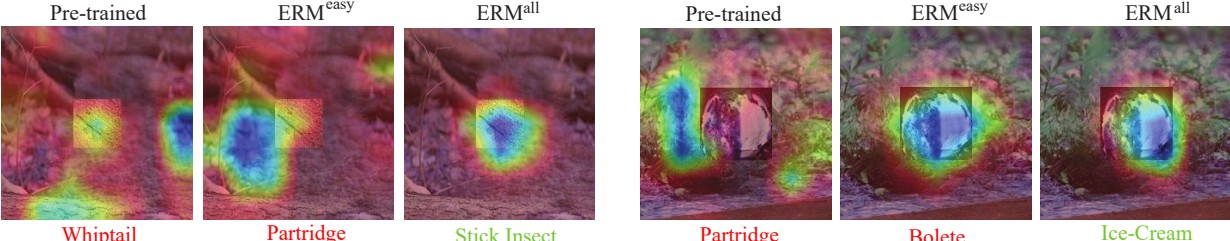

Figure 8: Gradcam visualizations showing regions of the image the model pays attention to in order to make the classification decision. Labels in red show false predictions and labels in green indicate correct prediction.

Table 6: Test Performance of different methods on Easy, Medium, and Hard categories in Hard-Spurious-ImageNet. Average accuracy is the average test performance of all the groups combined. The model is ResNet-50. Clean represents the performance of the model on clean ImageNet validation data. We see that there exists a trade-off between the clean accuracies and the performance on newly created dataset.

| Methods | Clean | Easy | Medium | Hard | Average |
|---|---|---|---|---|---|
| Pretrained | 81.21 | 65.39 | 48.50 | 16.54 | 43.48 |
| ERM | 65.78 | 74.84 | 66.67 | 57.56 | 65.94 |
| JTT | 40.06 | 60.90 | 53.09 | 46.49 | 53.50 |
| DFR | 69.25 | 72.47 | 65.65 | 59.79 | 65.97 |

Table 7: Breakdown of test accuracies with the ERM$^{\text{all}}$ model. The network architecture is ResNet-50. Compared to the overall trends shown in Figure 6, there is a notable improvement in the CoR and CeR groups, particularly at resolutions of $56^2$ and $84^2$.

| Size | CeO | CoO | CeR | CoR |
|---|---|---|---|---|
| $56^2$ | 62.25 | 60.8 | 54.56 | 54.45 |
| $84^2$ | 73.35 | 72.60 | 69.76 | 69.96 |
| $112^2$ | 77.19 | 77.13 | 75.34 | 75.48 |

In case of ERM, we also analyze the effect of the percentage of training data in minority groups, i.e. easy and hard groups, on model's test performance. We refer to ERM$_{\text{easy}}$ as the model that has been fine-tuned with data from the majority group only, i.e. 0% of data from medium and hard group. Conversely, we refer to ERM$_{\text{all}}$ as the model that has been fine-tuned with equal data points from all the groups, and ERM as the standard training data consisting of 20% of data from minority groups.

The results are depicted in the right plot in Figure 7. The x-axis shows the differently trained ERM models, i.e. ERM$_{\text{easy}}$, ERM$_{\text{all}}$, and standard ERM, and the y-axis shows performance across Easy, Medium, and Hard Groups. We see that training with the Easy group has worst performance on the Hard group. ERM$_{\text{all}}$ seems to narrow the gap between all groups. The accuracy of the Easy group remains similar across the models.

Table 7 presents the subgroup accuracy breakdown for the ERM$_{\text{all}}$ model. Compared to the overall trends shown in Figure 6, we observe a notable improvement in the CoR and CeR groups, particularly at resolutions of $56 \times 56$ and $84 \times 84$. Among all subgroups, the highest alignment with the clean accuracy for ResNet-50 occurs in the CeO group at a resolution of $112 \times 112$.

## 5.8 Analysing Classifications with Saliency Maps

We use Gradcam to visualize the predictions on the ResNet-50 model. Figure 8 shows the visualizations on the ImageNet-1k pretrained model and two variations of ERM: ERM$^{\text{all}}$ which is fine-tuned with equal data points from all the groups and ERM$^{\text{easy}}$ which is fine-tuned only with images from the Easy category, consisting of subgroups CeO and CoO for size $54 \times 54$ and $112 \times 112$ respectively.

The images on the left side of Figure 8 show a stick insect of size $56 \times 56$ placed in the center against an outdoor environment. The pre-trained and ERM$^{\text{easy}}$ model make their predictions by picking up cues from the backgrounds and predicting class *Whiptail* and *Partridge* respectively. Upon inspection, we find

that most of the images in these classes are set in similar environments, hence the model has learnt to associate the given outdoor environment with these classes and are ignoring the core features. ERM[all], however, is more robust to changes in environment and makes the correct prediction of class *Stick Insect*.

The images on the right show that, while the pre-trained model is confused by the spurious cues in the background, ERM[easy] makes the wrong predictions based on the cues in the core features and the background together. However, ERM[all] makes the correct prediction by mostly relying on core features. Figure 9 highlights the effect of the size of core features on the ability of the ERM[all] model to make correct predictions. Having a smaller core feature size results in the model incorrectly predicting class *Mushroom*.

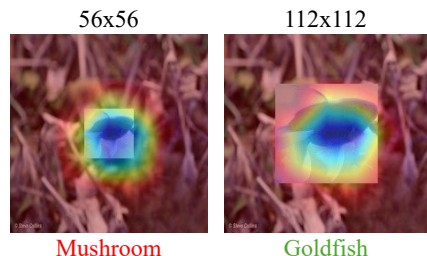

Figure 9: Effect of core feature size on model performance. Both the predictions are for the ERM[all] model.

## 6 Challenges and Future Work

The dataset variants of Hard-Spurious-ImageNet are proposed to understand the extent of background reliance as a function of size and location of core features. One of the limitations of the datasets is that they rely on ground truth bounding boxes of objects. In case of images where core features are not labeled by bounding boxes, no inpainting is performed on them, subsequently leading to core features in background and foreground occurring simultaneously. Moreover, the presence of secondary objects and clutter in the background makes it difficult for the models to learn small core feature sizes. The lack of segmentation bounding boxes for all images in ImageNet-1k restricted us to using object bounding boxes instead of masks. Currently, we have only experimented with one location per core object. For future work, we plan to experiment with different locations of core objects in the images and analyze the impact of using different network architectures with the dataset. Moreover, it would be interesting to extend this analysis to other datasets and models trained in different ways such as with contrastive learning, and various data augmentation techniques.

## 7 Conclusion

In this paper, we propose a variant of ImageNet-1k, Hard-Spurious-ImageNet, to help the deep learning community to better understand spurious feature reliance. We show that ImageNet-1k is center-biased and exhibits a bias towards large object sizes. We also provide an analysis showing that there exists a negative correlation between size and location of core features in an image and the strength of spurious cues in the background. To evaluate the robustness of models under these conditions, we experiment with a suite of group robustness methods. While some methods show marginal gains, none fully resolve the issue, highlighting the limitations of current approaches in settings where spurious correlations arise from dataset biases.

## Acknowledgments

This work was supported by the DFG Research Unit DFG-FOR 5336 "Learning to Sense". We used computing resources from the University of Mannheim and received support from the state of Baden-Württemberg through bwForCluster Helix and the German Research Foundation (DFG) under grant INST 35/1597-1 FUGG. Additional computing time was provided on the high-performance computer HoreKa at the National High-Performance Computing Center at KIT (NHR@KIT).

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

Table 8: Test Accuracies on Hard-Spurious-ImageNet-10 with highly spurious backgrounds.

| Model | Clean Accuracy | Object Resolution | Group Accuracies | | | |
|---|---|---|---|---|---|---|
| | | | CeO | CoO | CeR | CoR |
| Convnext-Base | 85.8 | $56^2$ | 57.4 | 38.6 | 8.2 | 1.0 |
| | | $84^2$ | 79.0 | 71.0 | 27.0 | 11.4 |
| | | $112^2$ | 83.2 | 79.8 | 45.4 | 17.0 |
| ResNet-50 | 82.2 | $56^2$ | 45.00 | 29.6 | 6.4 | 0.0 |
| | | $84^2$ | 70.8 | 58.4 | 17.0 | 9.8 |
| | | $112^2$ | 79.4 | 78.6 | 35.6 | 28.6 |
| CoATNet | 83.4 | $56^2$ | 20.9 | 21.8 | 0.8 | 0.0 |
| | | $84^2$ | 71.8 | 49.0 | 11.0 | 4.0 |
| | | $112^2$ | 75.40 | 80.8 | 18.2 | 23.0 |
| Hiera | 85.8 | $56^2$ | 46.8 | 22.8 | 1.0 | 0.0 |
| | | $84^2$ | 75.6 | 55.6 | 4.0 | 1.8 |
| | | $112^2$ | 78.6 | 74.6 | 16.0 | 10.6 |
| MVitv2 | 86.6 | $56^2$ | 29.0 | 15.0 | 0.4 | 0.0 |
| | | $84^2$ | 72.6 | 47.8 | 11.2 | 1.2 |
| | | $112^2$ | 72.4 | 66.0 | 18.4 | 12.6 |

## A    Benchmark Results

The results for Hard-Spurios-ImageNet-10 are given in 8. Similar to main paper, We test the performance of the datasets on 5 different pre-trained architectures: ConvNext-Base (Liu et al., 2022), ResNet-50 (He et al., 2016), CoATNet (Dai et al., 2021), Hiera-Base with MAE (Ryali et al., 2023), and MVit2-small (Li et al., 2022). All models are pretrained on ImageNet1k only. We see that Hard-Spurious-ImageNet-10 has far worse performance on groups CeR and CoR across all architectures and sizes. This indicates that the strength of spurious backgrounds is far greater than that of core features when the size of core features starts to decrease.

## B    Biases in ImageNet

Figure 11 shows the distribution of center and size scores for different classes in the training data of ImageNet. We calculate these scores using the available bounding boxes for ImageNet training data. Figure 3 refers to the distribution for the validation data.

## C    Inpaint Anything

The predicted masks from Segment Anything are dilated by a kernel size of 15 to avoid edge effects when the "hole" is filled by LaMa. Some examples of the inpainted data are given in Figure 10.

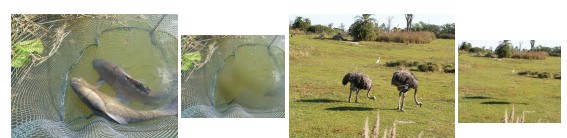
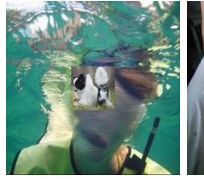
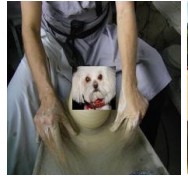
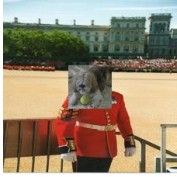

**Core:** Japanese Spaniel **Bg:** Snorkel     **Core**: Maltese Dog **Bg:** Potter's Wheel     **Core**: Shih-Tzu **Bg:** Busby Hat

Figure 10: **left**: Original images with their resized inpainted versions. **right**: Despite inpainting, the background (Bg) consists of cues that help the model predict the background label.

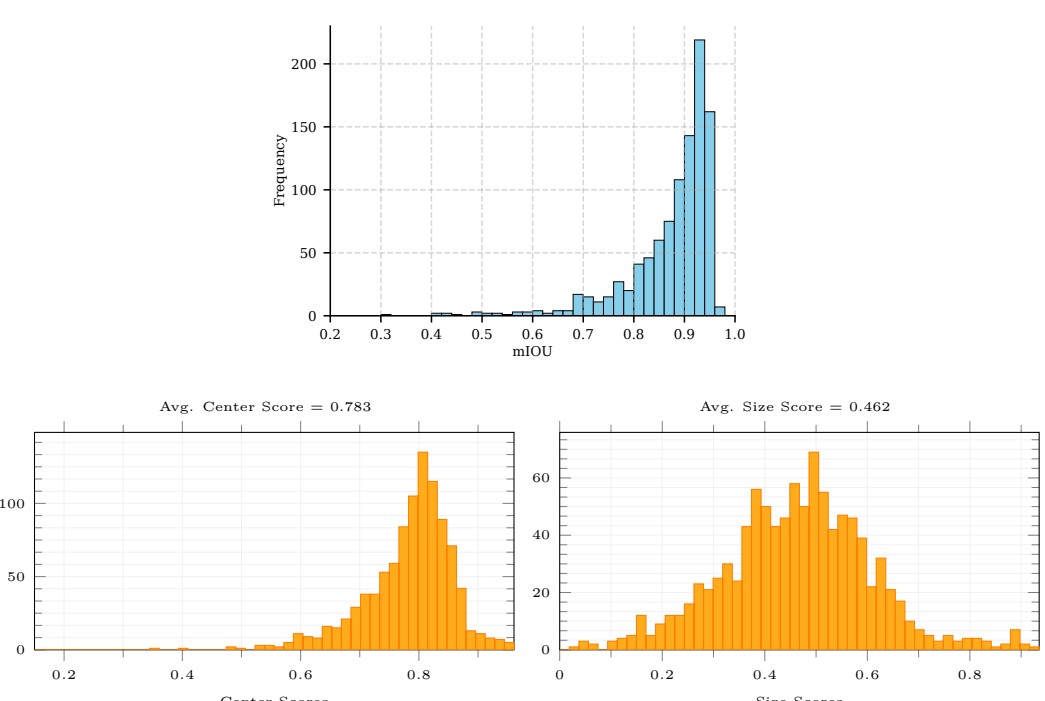

Figure 11: **(top)**: Class-wise mIOU scores between Grounding DINO predictions and ImageNet annotations on the validation set. Averaged mIOU is 0.875. **(bottom)**:Histograms showing distribution of scores in different classes of train data in ImageNet1k dataset. In the main paper, we show the distribution for validation data. This is the same plot for training data. The distributions are very similar.

## D   True Objects in Background

Ensuring that the backgrounds do not contain true objects depends on the fidelity of provided ImageNet annotations. We perform an additional analysis with a foundation model, Grounding DINO Liu et al. (2024), to extract bounding boxes from the images. We consider similarity scores between Grounding DINO predictions and the ImageNet annotations to analyze the correctness of ImageNet annotations. For ImageNet validation data, we get an overall mIOU of 0.8675 across all classes between both sets of bounding boxes with 139 classes having mIOU value less than 0.8 (see Figure 11 for a histogram by mIOU). This shows that the majority of the classes in ImageNet data have correct bounding boxes and the amount of objects from the foreground class in the background is negligible.

## E   Hard-Spurious-ImageNet with SAM

We also experiment with using the Segment Anything (Kirillov et al., 2023) model to obtain masks for the objects inside a bounding box and resize it to 3 different sizes (56, 84, and 112). The resized masks are then placed in the center and corner of the inpainted image, similar to the setting described in the main paper. At the moment, we only consider one object per image. Since we have access to ImageNet-annotated bounding boxes, we use them as prompts to be given to SAM. The results are shown in Table 9. Compared to the results in Table 1, the results with SAM are worse, mainly because the resized SAM object masks are not entirely accurate in cases where objects are small and thin, such as insects, etc. Hence, we prefer human-annotated ImageNet bounding boxes.

Table 9: Test Accuracies on Hard-Spurious-ImageNet with SAM Masks.

| Model | Clean Accuracy | Object Resolution | Group Accuracies | | | |
|---|---|---|---|---|---|---|
| | | | CeO | CoO | CeR | CoR |
| Convnext-Base | 84.43 | $56^2$ | 46.07 | 36.07 | 13.86 | 6.21 |
| | | $84^2$ | 61.18 | 53.92 | 31.04 | 22.30 |
| | | $112^2$ | 67.78 | 64.69 | 42.91 | 13.84 |
| ResNet-50 | 81.21 | $56^2$ | 29.33 | 24.34 | 6.68 | 4.36 |
| | | $84^2$ | 45.17 | 40.63 | 19.09 | 16.24 |
| | | $112^2$ | 55.24 | 52.56 | 31.34 | 29.87 |
| CoATNet | 82.39 | $56^2$ | 30.57 | 27.61 | 7.91 | 3.93 |
| | | $84^2$ | 50.94 | 44.66 | 21.03 | 15.63 |
| | | $112^2$ | 60.60 | 56.73 | 33.00 | 29.30 |
| MVit2 | 84.49 | $56^2$ | 37.94 | 25.88 | 9.08 | 2.92 |
| | | $84^2$ | 54.89 | 44.73 | 24.74 | 15.15 |
| | | $112^2$ | 63.73 | 57.94 | 36.80 | 30.60 |
| Hiera | 83.77 | $56^2$ | 39.88 | 27.06 | 10.34 | 3.198 |
| | | $84^2$ | 56.36 | 46.18 | 25.13 | 15.72 |
| | | $112^2$ | 66.14 | 60.38 | 39.15 | 31.63 |

## F   Group Robustness Methods

We use pretrained ResNet-50 trained on ImageNet1k for our experiments. The Base model is fine-tuned with batch size 256, constant learning rate of 0.001 for 20 epochs. The input images are randomly cropped with an aspect ratio in the bounds (0.75,1.33) and finally resized to $224 \times 224$. Horizontal flipping is applied afterward. A momentum of 0.9 and weight decay of 0.001 is used. For DFR, we normalize the embeddings using mean and standard deviation of validation data used to train the last layer, and use the same statistics to normalize embeddings of test data. We re-train the last layer for 1000 epochs, learning rate of 1, cosine learning rate scheduler and SGD optimizer with full-batch. We use $\ell_2$ regularization with $\lambda$ set to 100. These hyperparamters are similar to the ones set by Kirichenko et al. (2023) for optimizing the last layer for ImageNet-9 dataset (Xiao et al., 2021). Since, the data distribution in the proposed dataset and ImageNet-9 is similar, we assumed the same hyperparamteres. In case of JTT, models have the same hyperparameters as the ERM trained model. $\lambda_{up}$ is set to 50.

After extracting the embeddings from the pre-trained ERM model, the embeddings are normalized using **fit_transform()** and **transform()** functions of **sklearn.preprocessing.StandardScaler** for val and test data, respectively. For the JTT model, the images are applied with random resized cropping followed by horizontal flipping. No additional data augmentation is applied afterward. We also experimented with ConvNext-tiny pre-trained on ImageNet-22k and fine-tuned on ImageNet1k. We fine-tune the pre-trained model on the proposed data under various settings. ERM is trained by replicating the long-tailed distribution of the data, while ERM[easy] is trained only with the easy group. ERM[all] is trained with equal data points from all groups. DFR is trained by extracting embeddings from ERM, and re-training the last layer only. The number of train and test images is similar to the data setting described in the main paper.

## G   Dataset Details

Table 11 provides details on the Hard Spurious ImageNet dataset such as the available number of train, validation and test images per resolution and group.

Table 10: Test Performance of different methods on Easy, Medium, and Hard categories in Hard-Spurious-ImageNet. Average accuracy is the average test performance of all the groups combined. The model is Convnext-tiny.

| Methods | Easy | Medium | Hard | Average |
|---|---|---|---|---|
| Pretrained | 71.14 | 54.93 | 29.21 | 51.75 |
| ERM | 76.91 | 70.63 | 63.48 | 70.34 |
| ERM$^{\text{easy}}$ | 77.82 | 68.33 | 51.39 | 65.85 |
| DFR | 74.82 | 68.66 | 61.68 | 68.39 |

Table 11: Number of Images in every group of Hard Spurious ImageNet.

| Set | Resolution | CeO | CoO | CeR | CoR | Total |
|---|---|---|---|---|---|---|
| | 56 | 10k | 10k | 10k | 10k | 400k |
| Train | 84 | 80k | 80k | 10k | 10k | |
| | 112 | 80k | 80k | 10k | 10k | |
| | 56 | 50k | 50k | 50k | 50k | 600k |
| Test | 84 | 50k | 50k | 50k | 50k | |
| | 112 | 50k | 50k | 50k | 50k | |
| | 56 | 20k | 20k | 20k | 20k | 240k |
| Val | 84 | 20k | 20k | 20k | 20k | |
| | 112 | 20k | 20k | 20k | 20k | |

