# OpenReview forum: "Corner Cases: How Size and Position of Objects Challenge ImageNet-Trained Models"
_TMLR — Accepted by TMLR_

### Review · Reviewer_W8jg · 2025-05-15

**Summary Of Contributions:**

This paper focuses on biases caused by spurious correlations, particularly examining the impact of object size and position. Hard-Spurious-ImageNet is proposed, which contains images with various backgrounds, object positions, and object sizes. Experiments based on Hard-Spurious-ImageNet demonstrate that existing models exhibit significant biases toward object positions and sizes, and that ERM, DFR, and JTT methods fail to achieve considerable performance gains for worst-group accuracies.

**Audience:**

Yes

**Claims And Evidence:**

Yes

**Requested Changes:**

1. The object resizing approach fails to distinguish between genuine size bias effects and resolution-related artifacts, requiring clearer validation methods.
2. The current image synthesis method produces unnatural compositions. Advanced generative techniques (e.g., diffusion/GANs) should be considered to improve realism.
3. Expanding to state-of-the-art architectures and mitigation techniques would strengthen the findings.

**Strengths And Weaknesses:**

Strengths:
+ The study of biases related to object positions and sizes represents a significant research direction.
+ The paper is easy to follow.

Weakness：
- Regarding the object size bias study: The current methodology of simply resizing objects may confound size effects with resolution changes. How did the authors isolate and verify that the performance differences were indeed caused by size bias rather than resolution artifacts?
- As shown in Figure 5, the authors simply cropped objects and placed them arbitrarily in images without proper background blending, resulting in unnatural compositions. Why not employ more advanced techniques like diffusion models or GANs for image generation? Furthermore, given the significant divergence from real-world scenes in Hard-Spurious-ImageNet, can such testing truly reflect model performance on authentic images?
- The current evaluation appears limited in scope, employing only conventional models (e.g., ResNet50, ViT) and outdated debiasing approaches. We recommend incorporating more state-of-the-art architectures and mitigation methods to enhance the study's comprehensiveness.

---

> ### Author Response · Authors · 2025-06-14
> **Response to Reviewer Comments**
>
> Thank you for your valuable suggestions.
>
> 1. While diffusion models are capable of producing seamless background blending, we have chosen not to use them due to concerns about **domain shift** in the generated data. Recent studies (e.g., Yamaguchi and Fukuda, *"On the limitation of diffusion models for synthesizing training datasets"*, arXiv:2311.13090, 2023) have highlighted potential issues with using diffusion-generated data in downstream tasks, especially when models are pre-trained on natural datasets. Since the majority of pre-trained models we evaluate are trained on **ImageNet**, we have restricted our focus to this dataset to ensure compatibility and fairness in evaluation.
>
> 2. Additionally, we have now included results using **state-of-the-art vision-language models (VLMs)** and **vision transformers (ViTs)** in **Section 5** of the main paper.

---

> > ### Author Response · Authors · 2025-06-15
> > **Response to Reviewer Comment**
> >
> > 3. To address the resolution-related artifacts, we generate a new test set that contains objects resized by resizing the shortest side to 56, 84, or 112 and maintaining the aspect ratio. Moreover, we also put objects randomly on either of the four corners. Table1-5 in the main paper show results with both original and AR-preserving variant. The result indicate that when the aspect ratio is not preserved, it is harder to classify the "core" object.

---

### Review · Reviewer_1D6P · 2025-05-27

**Summary Of Contributions:**

The authors propose a new set of dataset that expose spurious correlations present in existing supervised ImageNet-trained Convnets. They first quantify a measure of how centered the data are based on the ground truth bounding box labels from the validation set of imagenet, as well as their bounding box ratio. Then, they create the so-called inpainted imagenet, a new dataset to show the bias toward object size and center using one model (ConvNext-Base [Liu et al. 2022] pre-trained on ImageNet22k). Based on that they introduce a series of new benchmarks with varying degrees of object’s ratio and position (CeO,CoO, CeR, CoR). Using inpainted imagenet, they measure the 10 classes with high acc thus spurious correlations and the 10 worst performing classes (“less-spurious”), where background features are not predictive of the sample’s accuracy. Then they create a dataset with 10 “less-spurious” classes of foreground objects and superimpose them with the 10 detected classes with the highly spurious backgrounds.

**Audience:**

Yes

**Claims And Evidence:**

Yes

**Requested Changes:**

A. From all the above, it is apparent that the submitted manuscript is more a first draft than a polished manuscript. The manuscript requires a major revision and it is far below the acceptance threshold in my view. I suggest the other start building meaningful comparisons that isolate only one factor. For instance if one wants to compare the model architecture (ViT, Convnext, ResNet) all models need to be trained on the same data, with the same objective  (supervised learning, self-supervised, weakly supervised, etc) and roughly have a similar number of hyperparameters. Same for all other factors. Right now there are multiple changes taking place simultaneously and it’s impossible to make a fair comparison. In general the most interesting parts of the analysis should be presented in the main manuscript.

B. The overall message of the manuscript gets too convoluted and it is hard to get a clear insight after reading it apart from “object size matters a lot to classification on supervised pre-trained convnets on imagenet”. It’s hard to discern anything else due to improper and unfair comparisons such as CLIP ViT, ConvNext and ResNets being pretrained on different datasets.

C. Further grouping of the groups to easy, medium and hard is not well motivated and further complicates the message.

D. A table summarizing the introduced datasets in terms of number of samples, object size , number of classes , would strongly ameliorate the paper’s readability.

E. The analysis includes many different model weights alternating from one section to another. I would be more interested to see which i) architectures, ii) pre-training strategies, iii) fine-tuning robustness methods are performing best under the most challenging benchmarks.

F. Gradcam visualizations are OK for convets, but if one wants to focus on interpretability the authors need to show how different models use regions of the image, including transformers (ViTs) using attention maps.


G. I personally like the inpainted ImageNet as a new idea/benchmark. Can the authors provide more comprehensive analysis on which models are the most sensitive? A scalability analysis could also be performed , i.e. start from CLIP Vit-B to ViT-G trained on LAION2B, do the larger scale models improve wrt spurious correlations? Similar experiments can be done with supervised models.

H. The top right corner is chosen while in principle any corner can be used. Can the authors modify the data to have a random corner? Fine-tuning on images that are either object-centered or the object is on the top right corner might lead to shortcut learning: it is sufficient to know when something appears on the corner.

I. I am strongly against the further grouping of easy, medium and hard. For instance, “Group CeO: Core object in the Center of image against its Original inpainted background” can directly be used as the easy benchmark without overcomplicated things, and CoR as the hard benchmark. I would include randomly positioning the object in one of the 4 corners as written above.

**Strengths And Weaknesses:**

### Strengths

Showing that the size of the object of interest strongly impacts classification accuracy, while impact from the object's location (center versus top right corner) is much less. Moreover, I personally like the inpainted ImageNet as a new idea/benchmark.

## Weaknesses (specific remarks)

1. The references are not correctly used: \citep{ref} can be used when the reference is not part of the sentence and must thus be in parenthesis. This makes the text hard to read.
2. A grammar spell check in the manuscript is highly advised.
3. New scores and figure 3-4 locations: why are the figures before the explanation of the scores?  For instance figure 3 is expected to be located close to sec 3.3.
4. “These scores are calculated by using Ground Truth bounding boxes of ImageNet.” This information must be stated in the beginning of Sec 3.1. As well as that the images are always resized to 224x224.
5. Figure 2. Right: why are the ticks ~100x100 ? Are the authors resizing the images? Does that change the overall qualitative assessment? If not, one can remove the x/y ticks.
6. Eq.1 : Why do the authors use the infinite norm?
7. Figure 5: The depicted abbreviations are not defined in the same page or in the image’s captions and thus are quite confusing to the reader.
8. Figure 3: what exactly is “inpainted ImageNet”? This could be solved by moving the image in its correct location and cross referencing in the main text where it is explained. At its current form it is particularly unclear what is happening. Right subfigure:The product of two scalar values between 0 and 1 may shift the distribution. Maybe averaging the two scores would have been a better alternative? In general, an intuitive interpretation is missing: for instance, classes with lower center/size scores are easier to classify while absent of semantic information.
9. What is the purpose of choosing ConvNext-Base pre-trained on ImageNet22k and fine-tuned on ImageNet1k? Why not pick a supervised or self-supervised model training only on ImageNet?
10. Figure 6: The figure can be improved by adding the clean accuracy (baseline as a horizontal line) or by adding the relative acc instead of having the separate table 1. Degradation on top of the bar. The coloring for Hard-Spurious-ImageNet versus the v2 can be replaced with a title on top of the 3 plots. The model names can be shown with a rotated text. The caption can be improved by explaining the abbreviations so that the reader does not need to see the manuscript. The position of the plot in the manuscript is expected to be close to sec 5.
11. ConvNext-Base trained on ImageNet22k. If the model is pre-trained on Imagenet-22k why do the authors claim that the outperformance is attributed to the “data augmentation pipeline”? Why did the authors use the ConvNext-Base trained on Imagenet to have a fair comparison to the other weights?
12. Reference of Hiera-Base with Masked Autoencoder without reference or results is confusing. Either add the results on table 2 or remove it. Paper ref. Is missing so the reader doesn't know which model you are referring to.
13. I don't understand what fig 7 is showing. Could the author provide an explanation?. The methods ERM, DFR, JTT need to be explained in more detail. I am not sure what ERM^{all} is supposed to mean. What does `conventional training to optimize average training accuracy` mean? If DFR trains only the last layer, why is it on the same plot as ERM in Fig. 7 , while the authors write `The Base model is fine-tuned with batch size 256, constant learning rate of 0.001 for 20 epochs`? What is ERM supposed to mean in the first place? Do the authors mean supervised fine-tuning the model on the new artificially superimposed training data? And if ERM is simply fine-tuning what is the point of presenting these alternative methods that do not show any significant improvement? And is the training stopped at 20 epochs since additional training is beneficial? This is all too opaque at the moment. I have no clue on what Figure 7 right is showing.
14. Sec 5.4: “We refer to ERM-easy as the model that has been fine-tuned with data from the majority group only i.e. 0% of data from medium and hard group.” Why is this in the experimental results? The so-called ERM easy medium and hard should be explained in Sec. 4 not Sec. 5.
15. Section 5 abruptly switches to a supervised trained resnet50 without any justification. Why did the authors choose the resnet50?
16. “The results in Table 3 show that pre-trained ImageNet models perform worst on the hard group.” If the hard group was meant to be the most difficult to classify correctly, why is that surprising?
17. The section “5.1 Effects of Data Augmentation and Self-Supervised Models” includes CLIP (i assume that’s what the authors mean by ViT-B in Tab. 2), which is not self-supervised.
18. Sec. 5: The majority and minority groups are not clearly explained until much later in the text.
19. Table 1 and Table 2: why are different feature extractors used? Why is the clean accuracy of the resnet50 not the same?
20. “we propose Hard-Spurious-ImageNet, a synthetic dataset” ← since multiple benchmarks are introduced this is not a single dataset but a set of benchmarks.

### Minor Weaknesses

- Please see into the syntax and typos such as `... by computing distinct scores, The detailed explanation of these scores are given in following sections.`

- Some sentences are hard to read: `We observed that the mask quality for some objects was not good enough, hence, we used provided bounding boxes for this work.`  Are you referring to the predicted masks from Segment Anything versus the ground truth? If yes, formulate it more clearly.  “we create four groups for the dataset as before” such formulations are unclear and make the manuscript hard to parse. “there is significant drop in performance compared with clean accuracies on standard validation dataset (see Figure 1).” ← this cross reference is probably wrong and table 1 is supposed to be referenced. “The Base model is fine-tuned” ← which model are you referring to?

---

> ### Author Response · Authors · 2025-06-14
> **Response to Reviewer Comments**
>
> Thank you for your valuable suggestions.
>
> 1. We agree that combining two scalar values (ranging between 0 and 1) via multiplication can shift the resulting distribution and potentially emphasize lower values more than an average would. Our rationale for using the **product** of the size and center scores was to capture their **interaction effect**. Specifically, a class is more likely to exhibit spurious behavior when **both** the object is small **and** positioned off-center. The product naturally emphasizes this joint degradation, whereas averaging would dilute this interaction. If the reviewer prefers, we are happy to provide comparative analysis figures for both the **average** and **product** of these scores.
>
> 2. We have revised our benchmark setup to improve clarity and consistency. We have removed results based on ConvNeXt models trained on ImageNet-22K and fine-tuned on ImageNet-1K, and now focus exclusively on **ConvNeXt models trained on ImageNet-1K**. This ensures consistent clean accuracies across the paper.
>
> 3. We have also expanded **Section 5.5** to provide more detail on the group robustness methods like ERM, DFR, and JTT. To clarify,  ERM_all refers to training with an equal number of data points from all defined groups, whereas the standard ERM refers to model training with the natural dataset having the long-tailed distribution nature.
>
> 4. All experiments from **Section 5** have also been run using the **ConvNeXt-Tiny** model, and the corresponding results are included in the supplementary material. If the reviewer finds it preferable, we are open to moving these results into the main paper.
>
> 5. Our grouping strategy is inspired by literature on **group robustness techniques**, which often consider **minority/majority** group splits (e.g., the Waterbirds dataset from Sagawa et al., *"Distributionally robust neural networks for group shifts"*, arXiv:1911.08731, cited by 2,119). Our prior analysis of ImageNet suggests that the dataset predominantly features large, centered objects—effectively forming a “majority group.” This motivates our categorization into **easy**, **medium**, and **hard** groups to reflect different levels of alignment with this majority structure. If this terminology causes confusion, we are happy to simplify the group naming to just **“majority”** and **“minority”**, or retain the original grouping scheme introduced earlier in the paper, depending on the reviewer’s preference.

---

> > ### Comment · Reviewer_1D6P · 2025-06-24
> > **clarification**
> >
> > In the new paragraph, `Furthermore, we also modify the test set by randomly positioning the object in the image...` the authors seem to have added the variant of placing the object in one of the four random corners. Where exactly are the results? Can the author please clarify this? Are all the results in the AR preserving column in all tables, such as table 1 based on randomly positioning the object? This is not clear to me.
> >
> > I mean the group names , Group CoR , and Group CoO refer to the top right corner as shown in the text page 6 below fig 6 (bullet points)
> >
> > Thanks.

---

> > > ### Author Response · Authors · 2025-06-25
> > > **Results for Random Location**
> > >
> > > Thank you for your response. In the AR preserving variant, two changes are made: 1. Random positioning of objects in any corner of the image. 2. Aspect ratio is preserved such that the smallest size is resized to 56, 84, and 112. The results are included in Tables 1 - 5 for both originally proposed data and the AR preserving variant. In the text in the bullet points, we refer to the top right corner for the originally proposed dataset (since the AR-variant is introduced later in the text). However, we have now added both the cases in the bullet points.

---

> > > > ### Comment · Reviewer_1D6P · 2025-06-25
> > > > **reply**
> > > >
> > > > Hello i have thought about it when I asked the question, and I was confused by ` 1. Random positioning of objects in any corner of the image.` <-- This applies only to the corner splits: Group CoO, Group CoR. So, calling it original and variant is not specific enough because:
> > > >
> > > > - `Group CeO`: Core object in the **Center** of the image against its Original inpainted background.
> > > > - `Group CeR`: Core object in the **Center** of the image against a Random inpainted background.
> > > >
> > > > Again, to the best of my understanding, these subsets are always centered (correct me if I am wrong), so their variant is only ` 2. Aspect ratio is preserved such that the smallest size is resized to 56, 84, and 112.`
> > > >
> > > > If the above are confirmed by the authors, could you reflect them to the paragraph starting with the sentence: `Furthermore, we also modify the test set by randomly positioning the object in the image`. Ideally, I would recommend revising the bullet points and the mentioned paragraph to make the creation of these splits clearer to the reader.
> > > >
> > > > I can understand that for the authors, these are obvious (what the authors refer to as `original` and `variant`); however, for the reader, it can become very confusing.
> > > >
> > > > I hope it helps.

---

> > > > > ### Comment · Reviewer_1D6P · 2025-06-25
> > > > > **Final remarks**
> > > > >
> > > > > - The sentence `Similar to the waterbirds dataset`, this subsection must be named `4.1 Hard-Spurious-ImageNet` for clarity and cross-referencing the subsection in fig 5.
> > > > > - Bold in the tables based on the points the author wants to emphasize (i.e. best performing variant) would help the readers.
> > > > > - Sticking to the existing literature's terminology, `minority/majority` instead of easy/hard, and removing medium would probably appeal to a broader audience. However, I can understand that would require lots of revisions all around the manuscript.
> > > > > - LAISON-400M --> LAION-400M
> > > > > - Tables 2 and 3 are more interesting compared to Table 4, so I would consider moving Table 4 to the supp. I am not sure if it improves the quality of the paper.
> > > > > - In Table 7, the authors present the result after fine-tuning with the artificial data. Are the authors using the same data for testing? And how much is the performance degrading on the original ImageNet validation set after fine-tuning with the synthetic generated data? This tradeoff should be shown. Please correct me if I got this wrong.

---

> > > > > > ### Author Response · Authors · 2025-06-26
> > > > > > **Reply**
> > > > > >
> > > > > > yes, your observation is correct regarding the grouping. We have now added an additional set of points to reflect the grouping in variant dataset.
> > > > > >
> > > > > > 1. We have renamed section 4 as Hard-Spurious-ImageNet.
> > > > > > 2. Made the numbers bold.
> > > > > > 3. Corrected the typo for LAION-400M
> > > > > > 4. Yes, the artificial data is used for testing in Table 7. We have now evaluated these models on clean validation set as well and included in paper. Indeed, a trade-off exists between the clean accuracies and the accuracies on the newly created dataset.

---

### Review · Reviewer_J4sv · 2025-05-31

**Summary Of Contributions:**

The manuscript analysis the impact of spurious correlations between foreground and background regions on image classification, in particular in connection with varying object sizes and positions. The primary finding is that when size and position deviates from the dataset biases, spurious correlations become a bigger problem for image classification in ImageNet-1K.

Method: The authors remove foreground objects from images that have annotated ground truth bounding boxes. They use an inpainting technique for this. The inpainted background-only image is now used as a canvas and a (new) foreground object is pasted on top at different positions and scales. The model, if it is confused by the background and the spurious correlation between the background and the foreground, will misclassify such images. The study shows that this behavior is high especially for small objects not in the center of the image.

**Audience:**

No

**Broader Impact Concerns:**

The manuscript addresses broader impact concerns with existing models by highlighting their biases. The manuscript itself does not raise new ethical concerns.

**Claims And Evidence:**

Yes

**Requested Changes:**

- To address the weaknesses a ViT and a VLM need to be included in most experiments in the manuscript.
- To address the issue with ambiguous images in the dataset because of the inpainting procedure, it is unclear what could be changed, but this would require rerunning all the experiments in any case.

One remedy for the last point that could work for instruction tuned large VLMs is that, one could detail in the prompt how these images were constructed and what exactly the model needs to focus on and to also include a few ambiguous examples as few shot prompts to clarify in the instruction. But I don't see an easy fix like this for standalone backbones like EVA-02 and ResNet50.

**Strengths And Weaknesses:**

### Strengths

- The manuscript revisits an important topic: model performance under spurious correlations and the impact of object size and position.
- The division into groups CeO, CoO, CeR, CoR makes the analysis easier to parse when compared to the scatter plots in Figure 3.
- The manuscript also studies the impact of group robustness methods like Deep Feature Reweighting and Just Train Twice in their setting.

### Weaknesses
 There are three main weaknesses
- The models explored are primarily ConvNet based whereas the computer vision community has wholesale converted to ViT based models. ViTs are analyzed only in a couple of tables (Table 2 and Table 7). A modern ViT needs to be included in all the Tables and plots for this manuscript to be relevant to the TMLR community. For example,EVA-02 from "Yuxin Fang, Quan Sun, Xinggang Wang, Tiejun Huang, Xinlong Wang, and Yue Cao. Eva-02: A visual representation for neon genesis. Image and Vision Computing" which can be obtained here for pytorch https://huggingface.co/timm/eva02_base_patch14_224.mim_in22k could be used as the main model for all experiments.
- Similar to the point above, a modern SoTA VLM such as ChatGPT / Gemini should also be evaluated. The prompt should list all the possible classes in ImageNet-1k and ask the chatbot to pick the right one for the image.
- The dataset construction process, as highlighted in figure 14, has the issue that despite in-painting after masking out the foreground object, the foreground object cues are still present in the image (fish contours, ostrich shadows, pottery wheel parts and pot [see also figure 5 top left where the yellow fish outline is visible and a blurry fish is present in the background]). Furthermore, Section 6 states that  "In case of images where core features are not labeled by bounding boxes,no inpainting is performed on them". Thus the resulting dataset has lots of ambiguous images. Under the multi-class classification setting used in ImageNet, the ground truth class is not known.

---

> ### Author Response · Authors · 2025-06-14
> **Response to Reviewer Comments**
>
> We thank the reviewer for their insightful comments, which have helped us improve the quality of the manuscript. We have carefully considered the suggestions and made appropriate revisions. Below, we address some of the specific concerns raised:
>
> 1. Due to the high cost associated with large vision-language models (VLMs) such as ChatGPT (e.g., GPT-4-Turbo costs \$0.01 per 1k tokens), and given that our dataset includes multiple variants and groups, running experiments across all configurations becomes expensive. Therefore, in this version, we report results using EVA-CLIP and CLIP models.
>
> 2. Regarding the inpainting process: removing the “core object” from an image can introduce ambiguity, as it may lead to the emergence of spurious cues. By subsequently adding the core object back onto the inpainted image, we create a meaningful test scenario in which the model must decide whether to rely on the spurious cue or the newly added core object. If we were to fully eliminate the ambiguity, the presence of spurious cues from other classes would be minimal. In that case, the experiment would primarily test the model’s sensitivity to object size and position. In contrast, our current setup is specifically designed to evaluate model performance in the **presence of spurious cues**, offering a more challenging and realistic test setting.

---

> > ### Author Response · Authors · 2025-06-15
> > **ViT results**
> >
> > We also thank the reviewer regarding the suggestion of reporting ViT results: We report results using MVITv2 (Li et al. (2022)) in table 1. Further, the new sections 5.2 and 5.3 are dedicated to analyzing the scaling behavior of ViTs and the effect of training strategies.
> > We hope this addresses the reviewers suggestion. And are happy to discuss or add further aspects.

---

> > ### Comment · Reviewer_J4sv · 2025-06-16
> >
> > The cues left behind after attempted removal of the 'core object' are not spurious as pointed in the examples in my review. The test scenario is therefore one where genuine evidence exists for both core objects and the model needs to realise that the sub-window patched on top needs to be prioritised without the model being trained or instructed to do so.

---

> > > ### Author Response · Authors · 2025-06-17
> > > **hand-picked failure cases**
> > >
> > > Dear reviewer,
> > > thank you for opening this point of discussion. As can be seen from the caption, the figure (now Figure 10) shows extreme and hand selected failure cases, which we show in order to discuss rare and unfortunate events that can not be entirely avoided. In accordance with your comment, we also discuss this issue in "Challenges and Future Work". However, we need to point out that these events are indeed rare and upon manual inspection, we can assure that most bounding boxes provided in ImageNet are indeed correct.

---

### Decision · Action_Editor_ijJv · 2025-07-20

**Recommendation:** Accept with minor revision

**Additional Comments:**

During the author-reviewer discussion period, the authors revised their paper to address reviewers' concerns. Most of the reviewers agreed that the current manuscript marginally exceeds the bar of acceptance, but still, there is a gap to be improved. I request a minor revision, mostly about the manuscript.

**Major: Need more discussion with the previous works**

The idea, altering the background to examine the robustness is also done by Xiao et al. (2020). Although the method is mentioned in 2.1 very briefly, it is better to clarify what is the main difference between Xiao et al. and this paper. For example, Xiao et al. used a limited number of classes, as this paper mentions, and this paper focuses on the crop sizes. Also, Xiao et al. zeroed out the object bounding boxes to make the background, but this paper fills the boxes using an inpainting model. However, I think that we need to give more credit to Xiao et al. in terms of the background alternation idea. The AC strongly suggests improving Sec 2.1. in terms of the technical difference between the proposed dataset and the existing datasets. The current version only mentions that there were some datasets, but does not clearly explain the difference between previous works and the proposed one, as well as the contribution of this paper.

- Xiao et al., Noise or Signal: The Role of Image Backgrounds in Object Recognition

**Major: Missing citations**

When the AC checked the paper, many methods were mentioned without proper citations. These methods should be cited in the camera-ready version.

- ImageNet-A (Sec 2.3)
- ObjectNet (Sec 2.3)
- LAION-400M (Sec 5.1)
- ViT (Sec 5.2)
- DeiT (Sec 5.3)
- AutoAugment (Sec 5.4)
- Random erase (Sec 5.4)
- Mixup (Sec 5.4)
- CutMix (Sec 5.4)

**Major: Citation formatting**

Despite Reviewer 1D6P's comment, some citations are still not properly formatted. For example,

> ConvNext-Base Liu et al. (2022), ResNet-50 He et al. (2016), CoATNet Dai et al. (2021), Hiera-Base with
MAE Ryali et al. (2023), and MVit2-small Li et al. (2022).

(Sec 4, pg 5)

>  Eva-Giant Sun et al. (2023) and CLIP Radford et al. (2021)

(Sec 5.1)

These citations should be `citep` rather than `cite` or `citet`. I found many similar improperly formatted citations. Please fix them all. I specifically found a lot of formatting errors in the revised parts and the Appendix. The authors should carefully review the paper to ensure correct and consistent formatting.

**Minor, but highly recommend updating**

- As far as the AC knows, ImageNet22k is not included in the first ImageNet paper (Deng et al. 2009) -- I think it was released in the Fall of 2011. Please consider citing the journal version (Russakovsky et al. 2015) for 21k or 22k (Sec 3.3 and Sec 5.2).
- Please use a consistent naming convention. E.g., ImageNet-22k vs. ImageNet22k, ImageNet-1k vs. ImageNet1k. Similarly, CLIP vs. Clip and Eva-Clip vs. Eva vs. EVA-CLIP ... (Sec 5.1), ResNet-50 vs. Resnet50, ... This can make the paper look unprofessional and low-quality.
- Formattings, such as "parameters i.e., 22 M" (it should be "parameters, i.e., 22M" (Sec 5.3)
- Please carefully review the manuscript in terms of formatting and fluency.

**Minor, maybe it is okay to not update, but if possible, I personally recommend updating**

- "SAM" is introduced without indicating that it means Segment Anything (Sec 3.3)
- In Sec 4., waterbirds dataset is introduced without any explanation. In the context of Sec 4, I think it is better to introduce how waterbird dataset is generated and what the dataset measures (distribution shift robustness) -- I think it looks okay to add an explanation in 2.2 or 2.3, but this is a mild suggestion.
- The name, Hard-Spurious-ImageNet-v2, could be confusing to readers who are familiar with ImageNet-v2 (Recht et al. 2019). But this is just a mild suggestion.
- Format error. Section F looks to have a format error. The newline between the first and the second paragraphs looks `\newline` between paragraphs (maybe not, but if yes, please fix it)

**Audience:**

Yes

**Audience Explanation:**

ImageNet is still a fundamental dataset to understand how deep neural networks work. I think the proposed problem is not very interesting and not very novel, but still worth exploring. For example, the core idea of the proposed dataset generation is already widely used by previous methods, such as background challenge (Xiao et al.) or waterbirds (Sagawa et al.). However, although this paper is not very novel and very interesting, I think the pitfalls of ImageNet classifiers are still worth exploring to improve our understanding of how DNN works.

**Claims And Evidence:**

Yes

**Claims Explanation:**

**Claim:** This paper claims that ImageNet-trained models can be biased to positional (location of the object within a given frame) and size (region-of-interest to image ratio) biases for different classes. To support this claim, this paper proposes a new synthetic ImageNet benchmark, named Hard-Spurious-ImageNet and Hard-Spurious-ImageNet-v2. Similar to Xiao et al, the proposed benchmarks are generated by altering the background of the ImageNet images. More specifically, the background images were generated by inpainting (while Xiao et al used unnatural image processing, such as zero-out or repeating pixels), and the resized object images (cropped from the original images) were pasted to the background images with various positions and crop sizes. This method can be a suboptimal design choice to examine the claim (especially, it contains an additional confounding factor, background), but I think it is an okay method.

**Evidence:** Figure 4 empirically shows the correlation between the validation accuracy and positions/sizes. Also, the experimental results show that the existing ImageNet models are not generalized well to the proposed dataset.

---

> ### Author Response · Authors · 2025-08-19
> **Revised Manuscript Updates**
>
> Dear AC,
>
> Thank you for your feedback. Please find below the changes we have made to the camera-ready version of our manuscript:
>
> 1. Introduced the contributions of Xiao et al. (2020) in detail in Sec. 2.2:
>    *Xiao et al., Noise or Signal: The Role of Image Backgrounds in Object Recognition*
>
> 2. Added the previously missing citations for:
>    - ImageNet-A (Sec. 2.3)
>    - ObjectNet (Sec. 2.3)
>    - LAION-400M (Sec. 5.1)
>    - ViT (Sec. 5.2)
>    - DeiT (Sec. 5.3)
>    - AutoAugment (Sec. 5.4)
>    - Random Erase (Sec. 5.4)
>    - Mixup (Sec. 5.4)
>    - CutMix (Sec. 5.4)
>
> 3. Corrected the usage of *citep* and *citet* throughout the text.
>
> 4. Added the citation for ImageNet-22k in Sec. 3.3:
>    *Olga Russakovsky, Jia Deng, Hao Su, Jonathan Krause, Sanjeev Satheesh, Sean Ma, Zhiheng Huang, Andrej Karpathy, Aditya Khosla, Michael Bernstein, Alexander C. Berg, and Li Fei-Fei. ImageNet Large Scale Visual Recognition Challenge. International Journal of Computer Vision (IJCV), 2015.*
>
> 5. Made naming conventions consistent across the manuscript (*ImageNet-22k, ImageNet-1k, ResNet-50, EVA, CLIP,* etc.).
>
> 6. Introduced SAM as *“Segment Anything”* in Sec. 3.3.
>
> 7. Added an explanation for the Waterbirds dataset (Sec. 2.2).
>
> 8. Changed the name *Hard-Spurious-ImageNet-v2* to *Hard-Spurious-ImageNet-10*.
>
> 9. Fixed spacing in Appendix Sec. F.
>
> We appreciate your guidance and hope these revisions address the concerns.
>
> Best regards,
> *Authors of Corner Cases*

---

> > ### Comment · Action_Editor_ijJv · 2025-08-19
> >
> > Thanks for the update! Everything looks great to me.
> >
> > I wonder if it is possible to append the Appendix to the main paper, as many other TMLR papers do (e.g., like the initial submission)
> >
> > If the authors are okay with the current version, I'll approve the camera-ready. If the authors prefer to have the Appendix and the main manuscript in the same document, please upload a merged one.
> >
> > Please let me know your preference!
> >
> > Best regards,
> >
> > AC

---

> > > ### Author Response · Authors · 2025-08-19
> > >
> > > Thank you for your response! We have now uploaded the merged camera-ready version.
> > >
> > > Please let us know if there are any further steps required.
> > >
> > > Best Regards,
> > >
> > > Authors of Corner Cases